# Benthic Biodiversity, Carbon Storage and the Potential for Increasing Negative Feedbacks on Climate Change in Shallow Waters of the Antarctic Peninsula

**DOI:** 10.3390/biology11020320

**Published:** 2022-02-17

**Authors:** Simon A. Morley, Terri A. Souster, Belinda J. Vause, Laura Gerrish, Lloyd S. Peck, David K. A. Barnes

**Affiliations:** 1British Antarctic Survey, High Cross, Madingley Road, Cambridge CB3 0ET, UK; terri.souster@uit.no (T.A.S.); belindavause@hotmail.com (B.J.V.); lauger@bas.ac.uk (L.G.); lspe@bas.ac.uk (L.S.P.); dkab@bas.ac.uk (D.K.A.B.); 2Faculty of Biosciences, Fisheries and Economics, Norges Arktisk Universitet, Hansine Hansens veg 18, 9019 Tromsø, Norway

**Keywords:** Antarctic, benthic blue carbon, carbon sequestration, cryosphere, climate change mitigation, benthic biodiversity

## Abstract

**Simple Summary:**

Seafloor biodiversity provides a key ecosystem service, as an efficient route for carbon to be removed from the atmosphere to become buried (long-term) in marine sediment. Protecting near intact ecosystems, particularly those that are hotspots of biodiversity, with high numbers of unique species (endemics), is increasingly being recognised as the best route to protect existing blue carbon. This study measured globally significant stocks of blue carbon held within both rocky (17.5 tonnes carbon km^−2^) and soft (4.1 t C km^−2^) substrata shallow (20 m) seafloor communities along the Antarctic Peninsula. Along the 7998 km of seasonally ice-free shoreline, 59% of known dive sites were classified as rocky and 12% as soft substratum. This gave estimates of 253k t C in animals and plants found at 20 m depth, with a potential sequestration of 4.5k t C year^−1^. More carbon was stored in assemblages with greater functional groups. Of the Antarctic Peninsula shore, 54% is still permanently ice covered, and so blue carbon ecosystem services are expected to more than double with continued climate warming. As one of the few increasing negative feedbacks against climate change, protecting seafloor communities around the Antarctic is expected to help tackle both the biodiversity and climate crises.

**Abstract:**

The importance of cold-water blue carbon as biological carbon pumps that sequester carbon into ocean sediments is now being realised. Most polar blue carbon research to date has focussed on deep water, yet the highest productivity is in the shallows. This study measured the functional biodiversity and carbon standing stock accumulated by shallow-water (<25 m) benthic assemblages on both hard and soft substrata on the Antarctic Peninsula (WAP, 67° S). Soft substrata benthic assemblages (391 ± 499 t C km^−2^) contained 60% less carbon than hard substrata benthic assemblages (648 ± 909). In situ observations of substrata by SCUBA divers provided estimates of 59% hard (4700 km) and 12% soft (960 km) substrata on seasonally ice-free shores of the Antarctic Peninsula, giving an estimate of 253,000 t C at 20 m depth, with a sequestration potential of ~4500 t C year^−1^. Currently, 54% of the shoreline is permanently ice covered and so climate-mediated ice loss along the Peninsula is predicted to more than double this carbon sink. The steep fjordic shorelines make these assemblages a globally important pathway to sequestration, acting as one of the few negative (mitigating) feedbacks to climate change. The proposed WAP marine protected area could safeguard this ecosystem service, helping to tackle the climate and biodiversity crises.

## 1. Introduction

The Anthropogenic release of carbon dioxide (CO_2_) is drastically altering the global carbon budget [1], with many consequences, particularly through its impact on increasing temperature and ocean acidification [2,3]. The losses of carbon-rich habitats, such as forest [4], wetlands [5], kelp forest [6], coral reefs [7], salt marsh [8], mangroves [9] and sea grass beds [10], is exacerbating this increase, through the loss of major biological stores of carbon. In contrast, polar continental shelves are one of the few regions on Earth where there are increased opportunities for carbon storage [11]. Nature-based mitigation and adaptation solutions have, however, been undervalued and under-supported, especially in the ocean [12]. Identifying ways to conserve mature carbon stores (i.e., countering threats to existing near intact carbon-rich ecosystems) and identify any that have the potential to increase will be key to the optimisation of any potential mitigating action [12,13].

The importance of the blue carbon pathways that sequester carbon from sea floor biodiversity to burial into sediment is becoming increasingly clear [8,9,11,12,14]. Organisms that live on the sea floor (zoobenthos and algae) play a key role in the carbon pump as they either fix carbon from the water column (algae) consume, and recycle, carbon in their food, and CO_2_ from seawater, to build their body tissues and skeletons. They hold important stocks of ‘natural capital’, in the form of stored carbon, and due to their close proximity to deep water muds and silts likely have high sequestration efficiency (e.g., [11,14,15,16]). Coastal marine habitats tend to be sites of high carbon capture and, if, on death, there are suitable habitats close-by for long-term storage, they can provide efficient pathways towards sequestration. Seafloor biodiversity can, therefore, be a key (storage) link for converting carbon capture (photosynthesis, mainly by phytoplankton and macroalgae) to eventual sequestration in sediments [11]. This creates an urgent need to quantify the stocks and change in carbon capture, storage and sequestration across existing habitats, in order to understand threats and prioritise the protection of mature carbon stores. This is especially the case in the polar oceans where there is the potential for the protection of remaining near-intact systems and enhanced carbon uptake if such habitats are protected [12,17].

The ocean plays a key role in the Earth’s carbon cycle, absorbing nearly 30% of anthropogenic CO_2_ [18], with about half of this being absorbed by the Southern Ocean [19]. The ecosystem service of blue carbon storage could be enhanced as the Southern Ocean offers one of the few opportunities for mitigating climate feedback, due to changes in the cryosphere that are enhancing carbon drawdown [11]. In particular, the reduction in winter sea-ice duration has the potential to extend the summer season of primary production, the so-called “greening” of the ocean [17].

In addition to the enhancement of existing productivity, the collapse of ice shelves and the retreat of glaciers [20] has opened up new areas of ice-free ocean, which act as new carbon sinks [11]. There are approximately 240 ice-filled fjords on the Antarctic Peninsula, 90% of which have glaciers that are retreating [15]. These fjords are steep sided and sediment filled, increasing the probability that benthos living on the sides of these fjords will reach the sediment, increasing the chances of burial. This retreat also uncovers new areas of sea floor for colonisation by benthic zoobenthos and macroalgae, which store additional carbon within their tissues and skeletons [15]. Polar benthos tend to be long-lived [21] and the skeletons in their structures can accumulate considerable CaCO_3_, which can be buried when they die. Furthermore, such external skeletons are likely to hinder microbial breakdown (carbon cycling) of the organic carbon accumulated in soft tissues, the so-called immobilising of carbon in certain marine organisms [11].

In fjords, and at shelf depths, carbon standing stock and levels of sequestration are positively correlated with the functional diversity of the community, i.e., the number of different trophic groups, which is a measure of assemblage complexity [12]. The biodiversity of a few WAP rocky shores [22,23] and soft sediment shore assemblages in the shallows (e.g., at 20 m; [24,25]) have been described previously. However, the relationship between functional diversity, biomass and carbon standing stock is currently missing. Around the majority of the Antarctic the levels of biological carbon storage remain unknown, so large-scale change (in the only major region on Earth where research has shown it is likely to be increasing—but vulnerable) is currently unquantified [11]. To date, these estimates have rarely included standing stock and productivity values for macroalgae, which, although a small component of the benthic assemblage on the southern Antarctic Peninsula, becomes an increasingly important component further north [26,27,28]. While there are estimates of the rate at which land on the Antarctic Peninsula will become ice free [29] and historical satellite images have been used to estimate the rates of glacial retreat [20], there are still no reliable estimates of how much of ice-free coastlines will be rocky or soft sediments. This study aims to fill this gap to firstly estimate the relationship between functional group diversity and current standing stock of carbon on shallow rocky and soft substrata assemblages, and secondly, the additional carbon storage that could be realised if the Antarctic Peninsula coastline becomes ice free and acts as an increasing negative feedback to climate change.

## 2. Materials and Methods

### 2.1. Study Area

Biodiversity surveys were conducted in both the austral summer (January–March) and winter (June–October) at 3 sites adjacent to the British Antarctic Survey Research Station on Rothera Point at the southwest end of Adelaide Island, Western Antarctic Peninsula, 67°36′ S 68°08′ W (Figure 1). The rocky substratum study sampled sites adjacent to Cheshire Island in 2015 [14], while the soft substratum study sampled sites in North Cove and South Cove ([25]; between 2013 and 2015).

### 2.2. Study Design

Along three rocky substratum transects adjacent to Cheshire Island, SCUBA divers surveyed a minimum of three haphazardly allocated replicate 0.25 m^3^ quadrats at 6 m, 12 m and 20 m depth, in both summer and winter (Figure 1). Megafauna and colonised rocks were initially collected by hand before all visible remaining fauna within the quadrat were then collected using a battery-operated water pump suction sampler ([18]; Appendix A) and retained in a cylinder fitted with a 3 mm^2^ mesh collection bag. The methods for surveying soft substratum biodiversity, and the assemblage descriptions at 20 m depth only, are described in detail in Vause et al. [25]. In brief, samples were collected from soft sediment patches, larger than 3 × 3 m, at approximately 20 m depth. Megafauna and colonised rocks were initially collected as described above for the rocky substratum, before the remaining rocks, animals and sediment were sucked into a 1 mm mesh size sampling bag using an airlift (Appendix A).

All organisms were kept submerged in water during transport to the laboratory for analysis. Animals were sorted from any collected substratum and then identified using their morphological features and identification guides (e.g., Polychaeta, [30]; Mollusca, [31,32]; Echinodermata, [33]; Bryozoa [34]) by experts or preserved in 96% ethanol for later identification by DNA barcoding.

After identification, wet mass (blotted dry), dry mass (24 h at 60 °C) and ash-free dry mass (480 °C for 24 h) were measured for the majority of individuals. Mass of the remaining individuals was calculated from length mass relationships (Appendix A). For all but Hexactinellid sponges (which have a siliceous skeleton) ash-free dry mass (50% of organic mass) and ash mass (12% of skeleton) were converted into mass of carbon (following [35]. Each species was assigned a functional group (Table 1) to assess the relationship between numbers of functional groups and carbon standing stock. Two minor but additional functional groups were identified, autotrophs such as macroalgae and parasites (Table 1).

### 2.3. Modelling Shallow Water Substratum Types

Antarctic scientists and SCUBA professionals provided first-hand knowledge of locations and substratum types (hard, soft or mixed; Appendix A) at 20 m depth for dive sites along the Antarctic Peninsula. Dive sites that were adjacent to seasonally ice-free shores were mapped onto the coastline using QGIS, allowing us to estimate the proportion of seasonally ice-free shores that had hard, soft or mixed substrata. Distances for coastline that were either permanently ice covered or seasonally ice free in summer were calculated for areas of the Antarctic Peninsula that are free of permanent ice shelves, up to and including the South Shetland Islands (north of 69° S). Many permanent ice shelves are vast, with the shore often occurring many kilometres to landwards of the ice edge. These were therefore excluded from this analysis. A high-resolution coastline dataset was accessed from the SCAR Antarctic Digital Database (version 7.4—https://data.bas.ac.uk/items/e46be5bc-ef8e-4fd5-967b-92863fbe2835/; accessed 20 May 2021), which has attributes of ‘ice coastline’ and ‘rock coastline’. The dataset was simplified in ArcGIS software using a simplification tolerance of 30 m, which is the lowest spatial resolution that any lines should have been digitised from. Distances were then measured for each attribute using the South Pole Azimuthal Equidistant map projection. The proportion of known hard, soft and mixed substratum dive sites was scaled up to estimate the distance of the whole ice-free coastline that is expected to have rocky, mixed or soft substrata.

### 2.4. Estimating Assemblage Carbon Standing Stock and Sequestration

The carbon standing stock values determined in this study were scaled using the distance of rocky, mixed and soft substratum ice-free coastline. Various corrections from previous research in this area and local knowledge [35,36] were applied to these values. From diver experience it was estimated that the 20 m depth zone stretches, on average, 40 m perpendicular to the shore [37]. The raw values were therefore multiplied by a factor of 40 to convert our 1 m strip around the Antarctic Peninsula into a 40 m-thick strip, which we estimate contains the 20 ± 4 m isobath. A recent regional study found that the carbon content of infauna and epifauna (combined) in fjord floor sediments was approximately one fifth of the total particulate organic carbon [15]. While similar studies have not been completed for rocky substratum, the export of carbon is expected to be higher from rocky shore assemblages (because of typically higher production and current velocity, see [36]). Much of this rocky shore productivity is expected to be exported down steep slopes to deep sediments, a clear route to sequestration, and so all carbon standing stock values were multiplied by 5.

Macroalgae forms a small component of benthic assemblages in Marguerite Bay [26], due to the frequency of ice scour in the shallows, but becomes increasingly important further north on the Antarctic Peninsula. Macroalgal standing stock biomass was taken from the literature for a mid-point along the WAP (64° S, Anvers Island; [34]). This wet mass was converted to dry mass (12.7% of wet mass) and carbon content was estimated at 30% of dry mass (see [38] for conversion factors). Approximately four times the standing stock of global kelp forests is exported [39] and so these values were multiplied by 5 (=1 × stock + 4 × export). Productivity, in terms of net carbon fixation, was calculated using the values from Antarctic macroalgae summarised in Runcie and Riddle [40]; values of 2 g C.kg wet mass^−1^ h^−1^ were used to calculate the annual productivity of this mass of macroalgae. Between 2 and 9% of macroalgal carbon is typically sequestered into sediments [41,42] and so an average value of 6% was used to calculate the proportion of carbon likely sequestered from macroalgal productivity. The standing stock and sequestered carbon values were converted into CO_2_ equivalents by multiplying by a factor of 3.67 (molecular mass of CO_2_ is 3.67 × that of carbon).

### 2.5. Statistical Analysis

Preliminary analysis found no significant differences between replicate transects within the sites or between replicate samples from each depth and so these were pooled throughout the subsequent analyses. Prior to analysis data were tested for normality using Anderson–Darling tests and for heterogeneity of variance using Levene’s test. Species richness was normally distributed, but faunal density and biomass on rocky substrata were normalised through log10 transformation. Data were analysed using ANOVA (GLM, MINITAB version 17 for windows).

ANOSIM, SIMPER and nMDS analysis was completed in Primer 7, to identify the species driving the differences between organic carbon and skeletal carbon held in rocky and soft substratum assemblages. Shade plots indicated that fourth root transformations reduced the bias from the dominant functional group, sedentary suspension (SS).

The relationship between the number of functional groups and both organic and skeletal carbon standing stock biomass was investigated across substrata, depth and season using ANCOVA (GLM). The number of functional groups and biomass were both log10 transformed to normalize the residuals of the analysis.

## 3. Results

### 3.1. Hard Substrata Assemblage Structure

Species richness on rocky substrata was not significantly different between seasons (GLM: F _(1,56)_ = 1.55, *p* = 0.22; Figure 2A) and there was also no significant species interaction between season and depth (F _(2,56)_ = 0.91, *p* = 0.41). However, species richness significantly increased with depth (F _(2,56)_ = 69.64, *p* < 0.01). The highest mean recorded species richness, of 52 species m^−2^, was in the winter at 20 m. The lowest species richness was recorded in winter at 6 m, which was just 10 species m^−2^.

There were no significant differences between mean faunal densities (log10) on rocky substratum in either season (GLM: F _(1,56)_ = 0.73, *p* = 0.40; Figure 2B); however, there was a significant difference between depths (F _(2,56)_ = 16.28, *p* < 0.01). The greatest density was at transect one, 20 m depth in the summer, which was 2069 individuals m^−2^, of which 936 m^−2^ were spirorbid worms. Mollusca contributed to the highest density, followed by Annelida, with 18,376 and 8956 individuals present within the 57 m^2^ area sampled, respectively.

### 3.2. Comparisons of Biomass on Hard and Soft Substrata

There was a significant difference in rocky shore biomass (AFDM g) between depths (GLM: F _(2,56)_ = 26.76, *p* < 0.01) but no difference between seasons (F _(1,56)_ = 0.16, *p* = 0.69; Figure 3A). There was a significant difference in soft substratum biomass (AFDM g) between coves (F _(1,23)_ = 21.89, *p* < 0.01) but no difference between seasons (F _(1,23)_ = 1.93, *p* = 0.18; Figure 3B).

### 3.3. Faunal Assemblage Comparisons

Echinodermata had the highest biomass of both rocky and soft assemblages but Mollusca and Annelida were also high (Figure 4A,B). The maximum biomass of Echinodermata occurred on rocky shores (Figure 4A) and the highest biomass of Mollusca were in sediment (Hangar) assemblages (Figure 4B). The phyla Porifera (sponges) and Chordata (fish) were only found on rocky substrata and Priapulida (penis worms) were only found on soft substrata (Figure 4A,B). For both rocky and Hangar soft substrata assemblages, sedentary suspension feeders made up the highest biomass (Figure 4C,D). In soft substratum assemblages, grazers and flexible feeders made up the next biggest portion of the biomass (Figure 4D). In contrast, biomass was more evenly spread across functional groups in rocky assemblages.

### 3.4. Functional Groups and Carbon Standing Stock

The organic (R = 0.62, *p* < 0.01; Appendix A) and skeletal carbon standing stocks (R = 0.59, *p* < 0.01; Appendix A) within functional groups were significantly different at the three depths, 6, 12 and 20 m on the rocky shore. The mobile hard-shelled scavenger/predators functional group (PL) was the main difference in both organic (21.7%) and skeletal (19.7%) carbon standing stock between 6 and 12 m (Table 2). Sedentary suspension feeders (SS) for organic (16.2%) and flexible feeders for skeletal (16.8%) carbon were the functional groups that were the most different between 6 and 20 m (Table 2). SS was the main functional group difference in both organic (18.3%) and skeletal (18.3%) carbon between 12 and 20 m (Table 2).

The organic (R = 0.86, *p* < 0.01; Appendix A) and skeletal carbon standing stocks (R = 0.93, *p* < 0.01; Appendix A) within functional groups were also significantly different between Cheshire (rocky substratum), Hangar and South Cove (both soft substratum) at 20 m. The biggest functional group difference between coves was for sedentary suspension feeders (SS), and this difference was between the soft substratum assemblages in Hangar and South Cove (organic, 33.8%; skeletal, 32.1%; Table 3). The biggest rocky to soft substratum functional group difference was in mobile hard-shelled scavenger/predators (PL) between Hangar cove and Cheshire skeletal carbon (20.2%; Table 3).

The number of functional groups varied from 3 to 12 in each replicate quadrat (Figure 5). Both organic and skeletal carbon stocks significantly increased with the number of functional groups (organic carbon, R^2^ = 0.45, skeletal carbon R^2^ = 0.66; Appendix A; Figure 5). For organic carbon, this relationship did not vary with season, depth or substratum type (all *p*-values > 0.05; Appendix A), whereas substrata was the only factor influencing the skeletal carbon relationship with functional group, with higher skeletal carbon associated with rocky substrata (F _(1,80)_ = 45.61, *p* < 0.01; Appendix A).

### 3.5. Estimating Shallow Benthic Carbon Stores

The standing stock of particulate organic carbon in the benthic assemblages ranged from 942 t C km^−2^ for hard substrata to 391 t C km^−2^ (Table 4) for soft substrata. There was a high variability between measurements, with a coefficient of variation of 140%. The standing stock of macroalgae was approximately 30% of the total (294 t C km^−2^ ± 61%) on rocky substrata. The annual productivity of benthic assemblages (excluding macroalgae) ranged from 136 t C km^−2^ year^−1^ for hard substrata to 82 t C km^−2^ y^−1^ for soft substrata. Macroalgal productivity was highest at 179 t C km^−2^ y^−1^ for hard substrata. Estimates for ultimate sequestration were 6.8 and 4.1 t C km^−2^ for hard and soft substrata benthic assemblages, respectively. Sequestration estimates from macroalgae were more than 50% higher for rocky substrata assemblages (10.7 t C km^−2^ y^−1^) than from other benthic sources, resulting in a total of 17.5 t C km^−2^ y^−1^ estimated for rocky substrata and 4.1 t C km^−2^ y^−1^ for soft substrata.

Analysis of the coastline between permanent ice shelves along the Antarctic Peninsula and islands North of 69° S showed that 7998 km was ice free (Figure 6). The majority of the ice-free coast was fragmented, surrounded by ice-covered coastline. On ice-free sites, 59% of dive sites (127) were classified as rocky substratum, 29% (61) as mixed substratum, and 12% (26) as soft substratum (Figure 6). These proportions were scaled up by the length of ice-free coastline to give 4746 km of rocky, 2280 km of mixed and 972 km of soft substratum. Calculations suggested that there was a total standing stock of 253,000 tonnes of carbon (Table 5), with most of this found on hard rather than soft substrata. The productivity of such standing stock led to a calculated sequestered total of 4500 t C y^−1^, or 16,400 t CO_2_e y^−1^.

## 4. Discussion

Southern polar benthic communities largely consist of ‘intact’ habitats, comprising high proportions of endemic species, in carbon-rich habitats which meet many of the key criteria for top priority protection, addressing both climate change and nature loss emergencies [12]. This region has some of the largest natural negative feedbacks on climate change and considerable societal value that has been little considered until recently [11,13,15,43]. Our findings suggest that there is a large potential for carbon sequestration in the coastal shallows of the Antarctic Peninsula, which have not previously been explored on this scale. In this study, the estimated sequestration potential in benthic assemblages along the ice-free shores of the Antarctic Peninsula (17.5 t C km^−2^ for rocky substratum and 4.1 t C km^−2^ for soft substratum at 20 m) were higher than previous estimates for the shelf (1.9 t C km^−2^, [43]). The addition of macroalgae to the rocky substrata assemblage is responsible for much of this increase, along with the inclusion of infauna. The rocky substratum site measured in this study (Cheshire Island) was also steeper than the WAP site (South Cove) measured by Barnes [43], offering greater protection to seafloor assemblages from iceberg scour and, therefore, higher carbon standing stock. On these steeper shores ice scour disturbance is likely to lead to a significant increase in the export of carbon, close to deep-water sediments where the potential for burial and ultimately sequestration is high [44]. Frequent iceberg scour may have ‘disguised’ the productivity of Antarctica’s shallows (0–50 m), with the potential to immobilize ten times as much carbon compared with deeper waters [43].

Such values for carbon sequestration are one or two orders of magnitude less than that measured in mangroves (174 g C m^−2^; [45]) although recent studies, including the current one, are finding that the potential carbon transfer to the sediment has been underestimated (fjords; [15]). The Antarctic coastline has few, if any, high-carbon-accumulating habitats, such as wetlands and mangroves, as the majority of the coast is exposed to wind and wave action. Around the globe these high-carbon-accumulating habitats are in serious decline through human disturbance, pollution and coastal erosion. The only low-wave-energy coastal habitats around the Antarctic are deglaciated fjords and bays, which are increasing in size and blue carbon sink extent due to glacial retreat [15,46].

In 2021, 54% of the coastline, 9404 km, was permanently ice covered. If the assemblages that develop on these coastlines are similar in nature to the current ice-free coastlines, then they could hold a further 298,000 t C standing stock, which could potentially sequester 19,400 t CO_2_e y^−1^. The WAP shelf area alone is large, extending over 806,000 km^−2^ [43], making it a globally important carbon sink. The whole of Antarctica has over 45,300 km of coastline [47] and if deglaciation processes extend beyond the Peninsula then the potential for very large C sequestration is high.

This is the first study to investigate the correlation between functional group biodiversity on blue carbon within both soft sediment and hard rock coastal benthic assemblages in the shallows (<25 m) around Antarctica. Previous studies have looked at sub tidal biodiversity on rocky shores [22,48] but infauna, found in pockets of sediment, were under represented due to constraints on sampling and carbon not being calculated. The use of underwater suction samplers (Appendix A) allowed a higher proportion of the total macrofaunal assemblage to be collected and identified, as it allowed the fauna to be collected from within these sediment pockets. This study enhanced our knowledge of biodiversity, species richness and how the functional group diversity of species impacts the carbon standing stock. Over a range of habitats surrounding the sub-Antarctic island of South Georgia, benthic carbon accumulation was higher in assemblages with a greater number of functional groups [15]. An assemblage with more functional groups likely reflects an increase in taxonomic richness with time since the last iceberg disturbance event, known to be one of the key drivers of taxonomic richness [49], resulting in a more complex and higher biomass community. This highlights the importance of a detailed understanding of coastal marine communities on the WAP, which is a hot spot for many aspects of climate change [2], and the need for improved baseline knowledge to better monitor responses to physical change. Resilience is linked to biodiversity within marine communities, with more biologically diverse communities being more likely to contain species with traits that will allow them to adapt to changing environments [50]. Protecting carbon-rich biodiversity clearly needs to be at the heart of blue carbon strategies [51].

We calculated Antarctica’s coastal shallows at 20 m depth to support standing stocks of 253k tonnes C, with sequestered carbon further estimated at 4k t C year^−1^ (Table 3). This is expected to increase in the near future in response to various aspects of climate warming, such as currently ice-covered shore line becoming ice free. Whilst the status of benthic assemblages on shores that are currently ice covered on the Antarctic Peninsula is largely unknown, the fact that 54% is permanently ice covered suggests that the current calculated figure would be more than doubled if a similar mix of rocky and soft substratum shore biota was established there. An extension of projected ice loss beyond the peninsula, to other regions of the Antarctic, would extend this further.

Ectotherm growth in the cold of the Southern Ocean is typically slow [21,52]; however, rapid colonisation and growth can occur when areas of the Southern Ocean lose their ice cover and become open water, promoting new areas for phytoplankton to bloom [53]. Ice-mediated impacts of climate change include the lengthening of the duration of the phytoplankton bloom, and therefore the length of the season for feeding and benthic productivity [43]. Results from the current study show no difference in biomass between summer and winter; however, a longer feeding period is expected to lead to increased growth each year, and greater carbon drawdown. Warming beyond the 1.5 °C warming target agreed under the Paris climate agreement [54] may threaten Antarctica’s vulnerable species which have so far lived within very narrow thermal bounds [2,55]. The impact of iceberg disturbance on the diversity within shallow-water communities has been studied in detail [43,49,56], with strong links detected between winter-sea ice duration and the levels of disturbance. These ‘positive’ aspects of climate change could be balanced by increases in iceberg disturbance due to the reduction in seasonal sea ice that locks icebergs in place (through winter–spring), at least until glaciers retreat beyond grounding lines and the iceberg numbers start to reduce. With current trajectories of disturbance, Barnes [43] showed that increased productivity due to the lengthening of phytoplankton blooms will likely outweigh losses due to increased disturbance from icebergs, ensuring the projected increase in this negative feedback.

Our study gives the first indication that the same relationship holds across a combination of soft and hard substratum assemblages in shallow waters. This is typically limited by ice scour [57], which maintains a pioneer-assemblage structure in the shallows [22,43]. The mechanism underlying this pattern is suggested as a combination of two factors, high levels of disturbance restricting species to all but the most disturbance-tolerant species, and low levels of disturbance allowing increased competitive exclusion, with an intermediate zone where both types can co-exist (Intermediate Disturbance Hypothesis, [58]. A recent study found that selective mechanisms, such as disturbance filtering and inter-species competition, reduce functional redundancy at the extremes of the measured disturbance gradient from 10 to 100 m on rocky shores [49]. Greater functional redundancy gives ecosystems resilience in the face of climate change, i.e., ecosystems are likely less affected by the loss of species if there are others than can still perform that functional role [59]. This redundancy can ensure that ecosystem services, such as blue carbon capture and sequestration, are able to continue [45], even if continued warming leads to the loss of some of the more vulnerable species [60,61].

Our scaling of measurements of Rothera assemblages to much of the Western Antarctic coastline, using substratum information from dive sites, involves considerable assumptions and unknowns. Due to considerations such as safety and dive objective, the dive sites included in this study were unlikely to be a random selection. Without detailed analysis of coastal topography, particularly under, or adjacent to ice-covered shores, it is difficult to test the assumption that dive sites on ice-covered versus ice-free shores will have similar substratum profiles. It could be argued that more exposed, steeper slopes would become ice free sooner than shallower and more sheltered coastlines, but concerted mapping efforts will be required before the estimates presented here can be improved upon. We also do not have information of how much of the sea floor adjacent to ice-covered coastlines is currently under ice at 20 m depth. Our study, however, provides the first attempt to assess how much extra blue carbon could be stored when currently ice-covered shores become ice free and new habitats become available.

The patchiness of shallow Antarctic marine communities caused by iceberg disturbance [62,63], results in a large variability in macro and mega benthos assemblage carbon, with a co-efficient of variation of 140% for rocky and 128% for soft substratum assemblages. The limited literature for macroalgae gave a co-efficient of variation of 62% [28,36], although it is likely that standing stock was underestimated, due to the size and patchiness of macroalgae. Applying benthic carbon data from one region (Marguerite Bay (67° S) for the assemblage and Anvers Island (64° S) for the macroalgae) to the whole of the Peninsula will clearly introduce errors. Assemblages change with depth, with half the rocky shore standing stock biomass at 12 m and a quarter of the biomass at 6 m, compared with the values measured at 20 m. Co-ordinated research effort is required between Antarctic researchers to improve the precision of these estimates. However, this study builds, and improves, on the coastal estimates of Zwerschke et al. [16] by considering wider environments beyond fjords.

For a variety of reasons, carbon capture to sequestration pathways at high latitudes and beyond national jurisdiction have been little considered by scientific study, climate mitigation planning or conservation. In some respects this is surprising, because it is likely to be amongst the largest ‘intact’ and least disturbed blue carbon habitats, as well as working as rare, negative feedbacks on climate change [11]. Furthermore, such coasts are often highly fjordic, which despite comprising just 0.3% of Earth’s surface, sequester 18% of marine sediment carbon [64]. More specifically, benthic assemblages along the Antarctic Peninsula seem to be a significant store of carbon and have the potential to sequester thousands of tonnes of carbon annually. Glacial retreat, receding ice shelves and primary production changes in response to sea ice losses along the Antarctic peninsula should all lead to an increase in this carbon sink. The additional carbon sinks expected as coastlines become ice free are expected to be even larger when fjords left by receding glaciers are taken into account [16]. Zwerschke et al. [16] recently reported carbon standing stocks within deglaciated fjords along the West Antarctic Peninsula and found that recent gains in carbon standing stock were much higher than previously estimated. Factors such as increased iceberg disturbance, and impacts of the loss of sea ice on pelagic fauna such as krill [65], may counteract some of this projected increase. However, designating marine protected areas on the Antarctic Peninsula is an easy win as they will protect one of the few negative feedbacks against climate change, at least at current temperature envelopes. The benthic fauna of the Antarctic Peninsula includes some of the most naturally disturbed but least anthropogenically disturbed assemblages in the ocean shallows (and perhaps least disturbed overall in deeper water); assemblages that have the capacity for increasing their current blue carbon stocks. The fauna has been globally recognised as important, due to its uniqueness (e.g., high proportion of endemics), and deserving of protection [12]. Protecting natural systems from anthropogenic disturbance is a much more effective way of protecting biodiversity and its associated ecosystem services than attempts to restore or recreate lost habitat ([66,67,68,69,70,71] but see [72]). Schemes that improve the strength of the carbon pump are not only key factors mitigating climate change but are key policy tools in the drive towards “net zero” economies. However, one of many challenges will be how to energise nations to consider areas beyond national jurisdiction, given that they would not count towards any one country’s nationally determined contributions [73,74,75]. At COP and other climate meetings there is considerable talk about solving a ‘global problem’, yet many governments seem more concerned with the appearance of progression towards ‘net zero’ rather than actual progress (for example by ignoring consumption emissions). There is a multitude of negative effects of climate change, but this paper highlights the positive counterbalancing effects from carbon storage within shallow benthic soft sediment and hard rock coastal communities and the potential for this carbon storage to increase, should this biodiversity be protected.

## 5. Conclusions

This study has estimated the blue carbon services provided by the shallow-water fauna and flora growing on both hard and soft substrata along the Antarctic Peninsula at 20 m depth. The proximity of deep-water sinks provides an efficient pathway from carbon standing stock to sequestration, resulting in long-term burial of carbon. With climate-change-mediated ice loss predicted to lead to a more than doubling of the ice-free shallow-water habitat, this environment provides one of the few global mitigating feedbacks against climate change. Protecting areas with high biodiversity, in near intact ecosystems, is highlighted as a key policy to mitigate both the climate and biodiversity loss crises.

## Figures and Tables

**Figure 1 biology-11-00320-f001:**
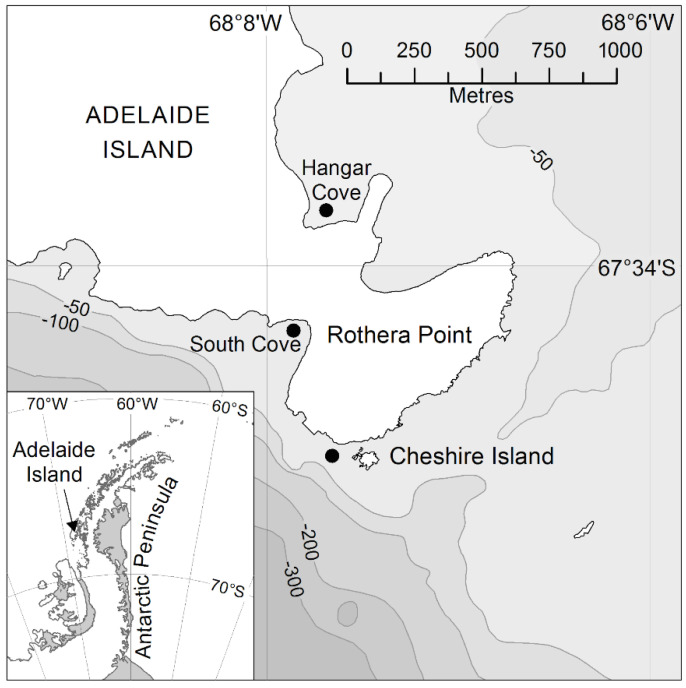
Sampling locations. Hard substratum was sampled from a site close to Cheshire Island. Soft substratum was sampled in Hangar Cove and South Cove.

**Figure 2 biology-11-00320-f002:**
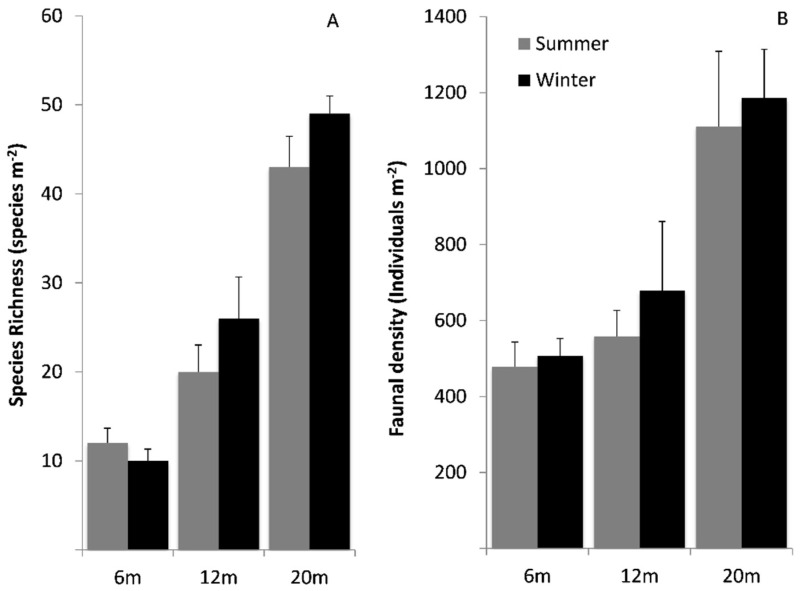
Rocky substratum assemblage at Cheshire Island, (**A**) species richness, (**B**) faunal density. Both summer and winter data are shown. Data are mean ± 1 SE.

**Figure 3 biology-11-00320-f003:**
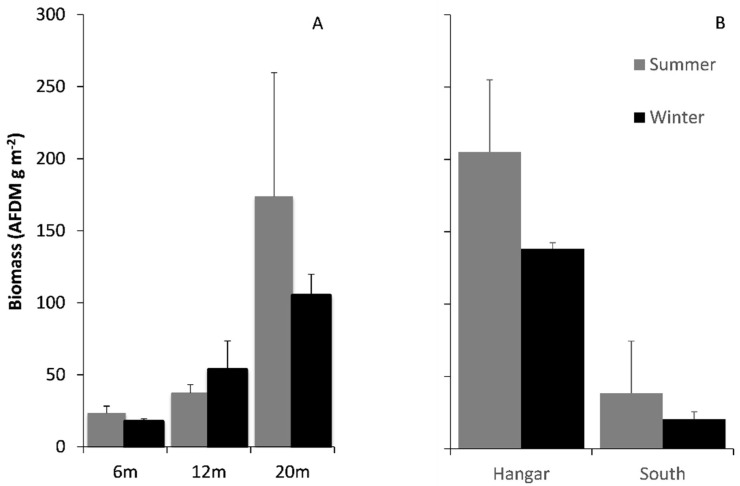
Comparison of faunal biomass, sampled on (**A**) rocky substrate at 6, 12 and 20 m depths. Additionally, (**B**) soft sediment in Hangar and South Cove was only measured at a depth of 20 m. Biomass in both summer and winter is shown. Data are mean ± 1 SE.

**Figure 4 biology-11-00320-f004:**
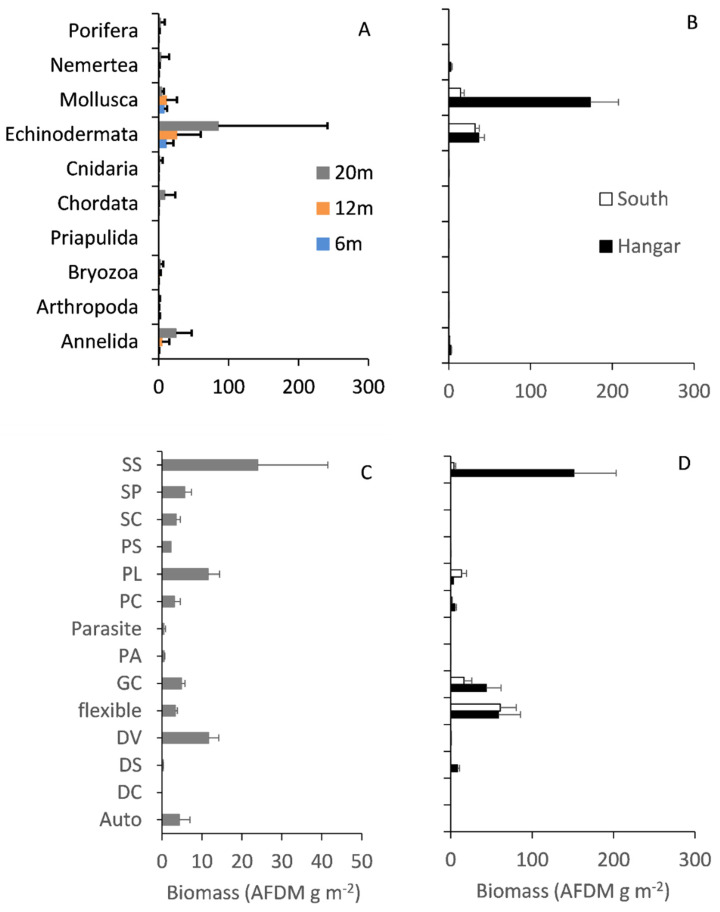
Biomass (AFDM g m^−2^) (**A**,**B**) within each phylum and (**C**,**D**) within each functional group. (**A**) Rocky substratum (Cheshire) at 6, 12 and 20 m deep, (**C**) rocky substratum at 20 m only. (**B**,**D**) Soft substratum (Hangar and South Cove) at 20 m depth. Data are mean ± 1 SE. Note that abscissae are logarithmic.

**Figure 5 biology-11-00320-f005:**
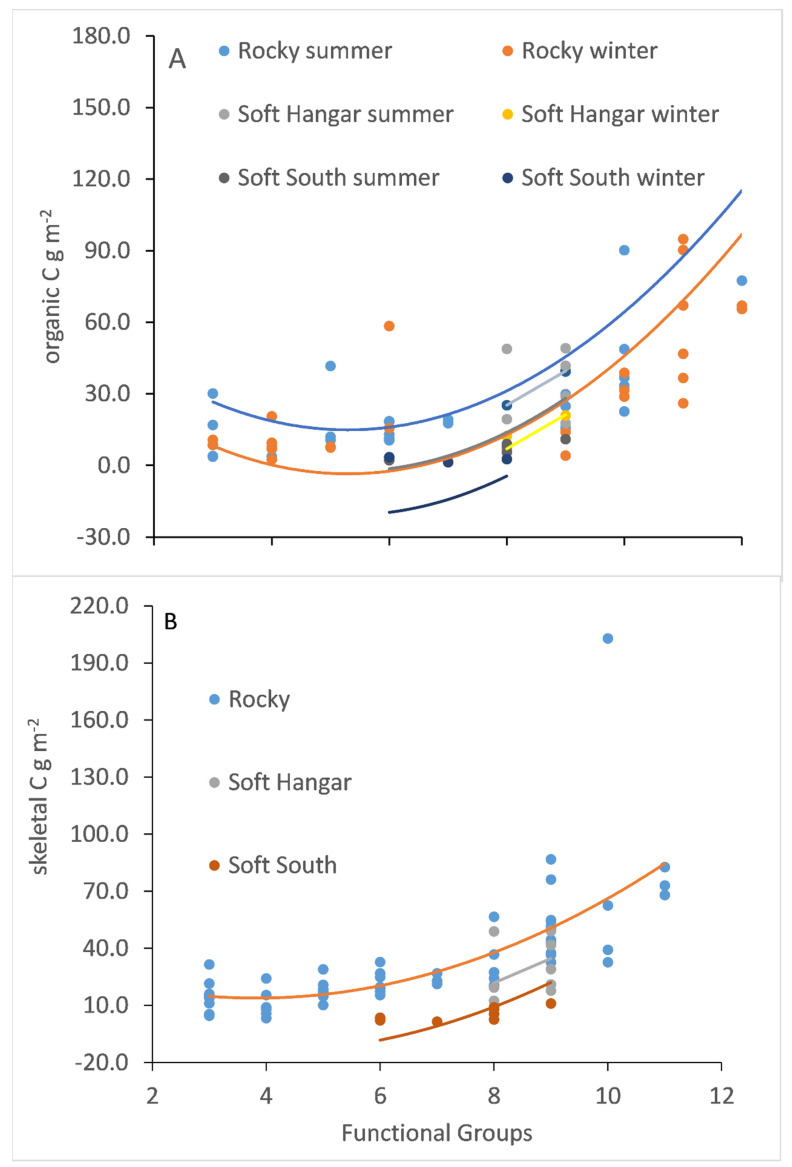
The relationship between (**A**) organic carbon and (**B**) skeletal carbon and the number of functional groups in the assemblage. Only substratum was a significant factor in the general linear model for skeletal carbon. S = summer, W = winter.

**Figure 6 biology-11-00320-f006:**
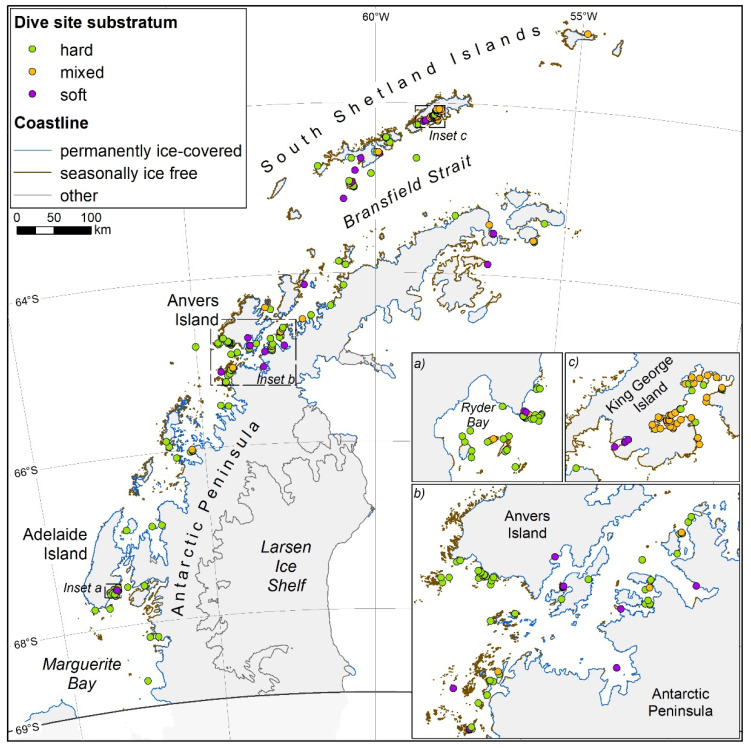
Geographic distribution of sites with known rocky (green), soft (purple) and mixed (yellow) substrata, with 3 of the best known locations highlighted. Estimates of the coastline between permanent ice shelves on the Antarctic Peninsula that are either seasonally ice free (brown) or permanently ice covered (blue). Unknown coastlines, hidden behind ice shelves, are indicated as other (grey). The coastline dataset was accessed from the SCAR Antarctic Digital Database (version 7.4—https://data.bas.ac.uk/items/e46be5bc-ef8e-4fd5-967b-92863fbe2835/; accessed 20 May 2021).

**Table 1 biology-11-00320-t001:** Abbreviations and description of the trophic guilds assigned to each species.

Code	Functional Group
SP	pioneer sessile suspension
SC	climax sessile suspension
SS	sedentary suspension
SM	mobile suspension
DC	deposit feeding crawlers
DV	deposit feeding sedentary (soft)
DS	deposit feeding sedentary (hard/shelled)
GC	grazer
PS	scavenger/predator—sessile soft
PC	scavenger/predator—sessile hard/shelled
PM	scavenger/predator—mobile soft
PL	scavenger/predator—mobile hard/shelled
PA	scavenger/predator—arthropod
Flexible	flexible
Auto	autotroph
Parasite	parasitic

**Table 2 biology-11-00320-t002:** SIMPER average % dissimilarity of the 4 functional groups driving most of the difference in organic and skeletal carbon between 6, 12 and 20 m on rocky substrata.

	Organic C			Skeletal C		
Functional Group	6 vs. 12 m	6 vs. 20 m	12 vs. 20 m	6 vs. 12 m	6 vs. 20 m	12 vs. 20 m
PL	21.7			19.7	12.9	
Flexible	14.1	12.7	9.5	18.8	16.8	12.5
PC	11.9			13.2		
DV	10.1	14.3	14.8	12.9		12.5
SS		16.2	18.3		16.2	18.3
SP		12.2	11.4			12.5
PA					12.7	

**Table 3 biology-11-00320-t003:** SIMPER average % dissimilarity of the 4 functional groups driving most of the difference in organic and skeletal carbon between Cheshire (rocky substrata) and Hangar and South Cove (both soft substrata) at 20 m depth.

	Organic C			Skeletal C		
Functional Group	Hangar vs. South	Hangar vs. Cheshire	South vs. Cheshire	Hangar vs. South	Hangar vs. Cheshire	South vs. Cheshire
SS	33.8	14.7	11.3	32.1		11.0
GC	12.7			15.4		11.8
Flexible	12.3	10.9				
PL	11.3	13.5	10.8	11.7	20.2	13.3
SP		12.6	15.0		11.7	
DV			13.4		11.6	
DS				16.6		
PA					12.7	11.8

**Table 4 biology-11-00320-t004:** Blue carbon in assemblages at 20 m depth on the Antarctic Peninsula. Standing stock of benthic invertebrates estimated in the current study. Literature values for macroalgae and conversion factors for annual productivity and sequestration are described in the text. Variability in the assemblage between replicates was scaled to assess the variability in carbon values.

	Substratum	Standing Stock t C km^−2^	Annual Productivity t C km^−2^ y^−1^	Sequestered t C km^−2^ y^−1^
Benthic organic carbon	Hard	648 ± 909	136	6.8
	Mixed	504 ± 667	106	5.3
	Soft	391 ± 499	82	4.1
Macroalgae	Hard	294 ± 181	179	10.7
	Mixed	147 ± 90	89	5.4
	Soft	-	-	-
TOTAL	Hard	942	315	17.5
	Mixed	651	171	10.7
	Soft	391	82	4.1

**Table 5 biology-11-00320-t005:** Estimated values for blue carbon at 20 m depth along the Antarctic Peninsula shoreline north of 69° S between permanent ice shelves. CO_2_ equivalents are also presented (see methods for details of calculations).

Substratum	Standing Stock t C	Productivity t C y^−1^	Sequestered t C y^−1^	Standing Stock t CO_2_e	Productivity t CO_2_e y^−1^	Sequestered t CO_2_e y^−1^
Hard	179k	60k	3.3k	656k	294k	12.2k
Mixed	59k	18k	1.0k	218k	65k	3.6k
Soft	15k	3k	0.2k	56k	12k	0.6k
TOTAL	253k	81k	4.5k	930k	371k	16.4k

## Data Availability

The data presented in this study are available at Morley: S., Souster, T., Gerrish, L., Vause, B., Peck, L., & Barnes, D. (2022). Blue carbon data for marine invertebrates living on soft substrata (20m South Cove and Hangar Cove) and Rocky substrata (Cheshire Island) around Rothera Point Antarctica (2013-2015). (Version 1.0) [Data set]. NERC EDS UK Polar Data Centre. https://data.bas.ac.uk/full-record.php?id=GB/NERC/BAS/PDC/01595, accessed 20 May 2021.

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
