# Peer review of "Benthic Biodiversity, Carbon Storage and the Potential for Increasing Negative Feedbacks on Climate Change in Shallow Waters of the Antarctic Peninsula"

_biology, 2022, doi:10.3390/biology11020320_

Round 1

Reviewer 1 Report

OVERALL COMMENT

The manuscript of Morley et al. titled “Benthic biodiversity, carbon storage and the potential for increasing negative feedbacks on climate change in shallow waters of the Antarctic Peninsula is interesting and has the potential to be a great publication. At it’s current version, I think there are still methods, analysis, and results which need further clarification and explanation, in order to support most of the discussion and claims made by the authors.

Rather than extending myself in an overall comment on all sections, I’ve opted for including a general comment, followed by specific comments, for each section.

Considering the work that still has to be put into improving the manuscript, and that these are analysis and result related, my overall recommendation is to reconsider the manuscript after major revisions are made.

ABSTRACTS AND INTRODUCTION:

General Comment:

I find the abstract to properly summarize the contents of the manuscript, there is the fact of the “mysterious” 18% more which needs further clarification in order to understand the potential doubling of blue carbon in the WAP.

In general, the introduction includes pertinent literature regarding zoobenthos and the state of blue carbon in the West Antarctic Peninsula sector of the Southern Ocean. One thing I am missing after reading the rest of the manuscript is mentioning the state of macroalgae. The sections Results and Discussion include and discuss data on macroalgal blue carbon, which is only mentioned before briefly in the Methods section of the manuscript, but appears absent in the Introduction. This leads me to the question, why include macroalgae later on, if not included in the background of your study? This represents one of the main topics to tackle and clarify for the Introduction, due to the important contribution of macroalgae to carbon stocks, productivity and sequestration show in the other sections of the manuscript.

Specific comments:

1- Line 22: In “18 % more of the… is still ice covered”. I find this rather confusing and roundabout, and it took me getting to the Results section to understand that you mean that from the WAP’s coastline there’s 18% more ice-covered than seasonally ice free (if I got this right). But if I take the numbers provided 7,998 km was ice free, whereas 9,404 km were permanently ice covered, which gives you (roughly) 46 and 54%, respectively. Wouldn’t it be better to be straight forward and provide the 54% value, rather than use the “18% more” (consider that if you compare the percentages, there’s an 8% difference, which makes everything even more confusing).

Using the 54% also fits better the proposed doubling of carbon, it is a lot more intuitive and easier to understand.

2- Line 33-34: Why do you provide distance instead of area for transforming from unit mass per unit area to stocks of carbon?

3- Line 36-37: As mentioned in comment 1, I recommend you replace the “18% more” by the actual percentage of coastline permanently ice covered.

4- Line 57-58: If the point is to introduce the term “zoobenthos”, I would recommend replacing “organisms” by “fauna”, since flora also has a benthic component, especially in the shallows (also shown in the results and discussion with the inclusion of macroalgae data).

5- Line 70: I think these is an extra “that”. Wouldn’t it make more sense to say “are protected” instead of “can be”.

6- Line 79 & 89: I suggest including a paragraph break here, to separate one idea per paragraph.

7- Line 84: For consistency, I would mention macrofauna as macro zoobenthos. This is considering the term previously introduced was “zoobenthos”. Same suggestion applies to “benthos” in Line 82.

MATERIAL AND METHODS:

General Comment:

The section has a few aspects for which I don’t understand the reasoning behind the choices made (see specific comments for details). My main concerns are: 1) the use of distance and a factor of 40 for extrapolating from biomass values, instead of defining clear areas; 2) the “late” inclusion of macroalgae (Line 171) in the manuscript, and; 3) that this data obtained from literature was used for a calculation included in the results. Regarding the later, I would argue that such a value, contributes a more to the discussion on benthic carbon stocks, production and sequestration, rather than to the zoobenthic topic the manuscript has (based on the background given in the Introduction).

Another important thing that needs to be clarified is how data was transformed. The first mentioning of data transformation implies natural logarithm, a few lines after, fourth root transformation is used, but the results are based on Log10 transformations. This needs consistency and serious correction/revision from the authors, since it is the foundation on which differences and conclusions are drawn and discussed.

Specific comments:

8- Line 140: While functional diversity is mentioned, no measure of it was calculated. We could argue that, at most, functional group diversity was used, and this is also hard to accept. What was considered, was the number of functional groups (the Log10 and the square Log10 of it, based in the text). Based on the work frame given for biological trait analysis in marine environments (e.g. Bremner et al. 2006, Beauchard et al. 2017, and literature therein), functional diversity is calculated based on functional traits rather than groups (the groups provided include more than one trait/modality). This is an aspect that requires work, especially to support statements such as “This is the first study to investigate the impact of functional biodiversity on blue carbon”, and the discussion based on the relation between functional diversity and blue carbon.

    • Bremner et al. (2006b) Methods for describing ecological functioning of marine benthic assemblages using biological trait analysis (BTA). https://doi.org/10.1016/j.ecolind.2005.08.026
    • Beauchard et al. (2017) The use of multiple biological traits in marine community evology and its potential in ecological indicator development. http://dx.dodi.org/10.1016/j.ecolind.2017.01.011

9- Line 154-156: Why not also estimate area, which is more straightforward to extrapolate biomass (mass per area) and abundance (individuals per area) data?

10- Line 161-163: If I understood correctly, you first multiplied your carbon estimates by distance a certain substrate covers, then this by 40, correct? Wouldn’t it be more straightforward to mention the 40m value was considered to estimate the area of a given substrate, and use this area to scale your carbon estimates?

Another way, a bit more time consuming, would be to use this 40m line and develop polygons in the GIS environment and measure their area. While time consuming, I could imagine the polygons might be reused by future studies.

Another aspect to consider is that you find clear differences in terms of biomass between your 6, 12, and 20m transects. If you only consider 20m values to estimate your stocks for the area covered by the distance a given substrate has times 40 (distance from shoreline to 20m depth, if I understood correctly), you are overestimating by a factor of 2 for the 12m depth line (almost half the biomass of the 20m transects) and by a factor of ~4 for the 6m depth line (roughly a fourth of the biomass of the 20m transects). This should be taken into account for all calculation. In case it was, this should be show in a crystal clear way in the text.

11- Line 168: There is a bracket missing after “[40]”.

12- Line 170: Could you elaborate the reason why you multiplied your values by 5?

13- Line 176-177: It’s not fully clear how the export of kelp forest standing stocks (4x) is related to multiplying your values 5x. Could you clarify the connection?

14- Line 179: Is there a “-1” too many? Or was there a “h-1” which was intended to be there and was removed?

15- Line 191-193: Could you specify which variables were normal and which weren’t?

16- Line 193: Does this mean that you had no zeroes in your data set? If there’s any, then rather than the natural logarithm (Ln(x)), the transformation should be Ln(x+1).

If you did use Ln(x+1) then please specify this. Another thing, if you used the Ln(x), why are values given as Log10(x)? These are not the same.

17- Line 195: Shouldn’t it be “fourth root”? Furthermore, didn’t you use Ln transformed data? If so, why do you now mention fourth root transformed data?

18- Line 196-198: Why did you transform the number of functional groups? Also, if here it is stated you used Ln(x), why is it that you use Log10(x) for your results?

RESULTS:

General Comment:

The first three subsections (based on headers, these are only two, since the first subsection has no header of its own) of the results appear as if written as the results from the analyses were read. While brings the important numbers straight to the point, the text is poorly connected and does not really highlight the findings you will discuss later on. This is not the case for the last subsection, in which the presentation of the estimations, which is better explained and presented. I ended up with more questions than answers, e.g.:

  • Why is there more data for the rocky substrate than for the soft sediments, when you measured exactly the same?
  • The macroalgae topic I mentioned in the general comment of the Introduction and Methods section.
  • Why not make a subsection for all topics? It would seem you’re missing a header (Line 200-230).
  • Why provide the nMDS when they do not bring much to the table, at least in terms of what you want to present?
  • Why use log transformed axis for your figures, when untransformed ones could show differences in a clearer way?

Specific comments:

19- Line 200-230 & Fig. 2: Why is no data or figure of this kind (diversity + abundance) given for soft sediments? As I understood it from the methods, also the one described in Vause et al. (2019), you should be able to provide these.

20- Line 200-300: I recommend reformat this part into 2-4 paragraphs. The first 1-2 on patterns in rocky substrata, stating no seasonal differences, but significant between depths differences. This can be complemented by which groups contributed most to abundance & biomass (these should be your SIMPER results). The next 1-2 paragraphs should contain the same, but for soft sediments.

21- Line 210: If you used Ln(x) transformations, why is it here mentioned as Log10(x)?

22- Fig. 3: The “parasite” and “auto” functional groups are not mentioned/included in Table 1. I think they should be included if considered for the figure. Also, since they appear to be new groups, mention why they were included.

Rather than using Log(x) scale, I recommend the use of untransformed scale to better represent the differences in terms of biomass for both, taxonomic units and functional groups.

23- Line 229-230: I would be careful when doing such statements using a Log(x) scale as reference. To better back this statement, I recommend to replot Fig. 3 using an untransformed scale.

24- Line 222-229: Considering you will be using data of Fig. 3C and D in the next subsection, I recommend you refrain from doing this here, and include this part in subsection 3.1.

25- Subsection 3.1.: Why are you providing comparison between transects and replicates, when you mentioned in line 187-189 that preliminary analyses showed no differences for part of your data? Why not also do the same for data presented in this subsection?

26- Fig. 4 & 5: In your case, the nMDS plots bring little to the table (you discuss carbon stocks rather than composition differences). The most important you draw from them in the text is which groups contribute to differences, for this, the SIMPER results outclass the nMDS plots. What you use from these plots can also be shown with a figure similar to Fig.3 but with skeletal and organic carbon values, and also making remarks in the text (partly done). If you want to keep the nMDS, I suggest to have them as supplemental material rather than in the main text.

Regarding captions: nMDS are a graphical representation of the among-samples resemblance pattern. They do not provide similarity between stations or factors, this is represented by a value which you usually calculate with a similarity analysis, SIMPER in the case of PRIMER. I think that providing a table does a better job if you want to show similarity percentages, significance of difference, and groups contributing to these differences. Furthermore, the nMDS plots are not a comparison, if your aim is to show this, Fig. 3 works better.

Is the “,m-2” supposed to be shown like this? I would assume it should have been “g C m-2” and in brackets.

27- Line 263-264: Can you express this in a more straightforward and intuitive way, using number of functional groups rather than using the roundabout “square of the number”.

28- Line 263-270: The paragraph is stiff and feels like reading the results of Table S3 as they show in the console. This isn’t inherently wrong, and my suggestion would be to clearly separate where you refer to organic carbon from the skeletal carbon results (maybe 2 short paragraphs in total), or better connect the sentences. E.g.: “Both organic and skeletal carbon stocks significantly increase with the number of functional groups. For organic carbon, this relation did not vary with season, depth or substrate (all p-values > 0.05). Whereas for skeletal carbon, substrata was the only factor influencing this, with a stronger relationship with number of functional groups for rocky substrate”.

One thing which is not clear to me is the differences in terms of number of functional groups. As I got it from the results, no data on the topic is provided, other than what is represented by Fig. 3. According to this, you basically should have all functional groups but DC for the rocky substrate, and all but SP, SC, Parasite, DC, and Auto for the soft substrate, correct? If this is the case, how was the GLM built? How is it that you have different number of functional groups? Could you provide a clarification, also in the Methods and Result sections

29- Line 278: Shouldn’t “co-efficient” be one word “coefficient”?

30- Line 282: Shouldn’t it be “highest”.

31- Line 283: What do you mean by “ultimate” sequestration? Is there a sequestration before?

32- Line 293: Shouldn’t it be “were” instead of “was”?

33- Line 299: There’s a space missing in between “Table” and “3”.

34- Line 303-307: The mention of A) and B) is not really reflected in the figure. What is shown in the Fig. but not explained in the caption are the locations a), b), and c). I recommend adding “(dots)” after sites.

Is “ice coastline” the same as “permanently ice-covered”, same for “ice-free” and “seasonally ice-free”, and what does “other” represent.

35- Line 308-313: Rather than results, I think this paragraph fits better as discussion.

DISCUSSION:

General Comment:

Overall, the physiological and development of MPAs topics are well discussed and supported. I do not see this for the passages relating functional diversity and blue carbon stocks, production and sequestration. This is mainly to the doubts present regarding, e.g. how the relation between number of functional groups and carbon was calculated; or why make claims based on functional diversity, when from a biological trait analysis perspective, you did not calculate this diversity, nor additional metrics such as redundancy and evenness.

An improvement of the results could indeed provide the necessary metrics to support the discussion of this manuscript, which at the moment, and to my opinion, are still lacking or not clearly shown.

Specific comments:

36- Line 318: There’s an “S” missing in “Southern”.

37- Line 319: Shouldn’t it be “meet” rather than “meets”?

38- Line 321: I think there’s a “the” missing between “some of” and “largest”.

39- Line 323-324: The fact that many areas of the Southern Ocean are unmapped (not only for blue carbon) is relatively known (also mentioned in part of the literature cited), this also makes logic rather than being surprising that very little is known regarding blue carbon in the shallows. With that in mind, I recommend removing this statement.

40- Line 328: You can remove “here” and “estimated”, since they both are mentioned at the beginning of the phrase “In this study, the estimated” (Line 326-327).

41- Line 328-329: I recommend to review these estimations based on comment 11.

42- Line 348-351: This passage is a bit long and complicated to follow. I suggest separate it into shorter and simpler sentences, e.g. “The WAP shelf area is large, extending over 806.000 km2. Here, the climate induced marine ice loss is resulting in an increased number of blue carbon habitats. Thus making/transforming the WAP a globally important carbon sink.”.

43- Line 352-359:

      1. You did not investigate the impact of functional diversity on blue carbon. Only that there is a relation between number of functional groups (square of the logarithm, not even number of) and blue carbon stocks.
      2. As for species diversity, there are indices for functional diversity (e.g. Bremner et al. 2006 and literature therein). As far as I know, most of these indices are based on individual traits per taxon, rather than on functional groups. A solution would be to estimate a tentative functional group diversity based on your functional groups (rough estimate of functional diversity since you have a mix of traits for most groups). Better, would be to separate traits for each functional group and from those calculate functional diversity (in a similar fashion as it was done by Vause et al. (2019)). Another would be to use a more detailed classification scheme similar to the one used by Robinson et al. (in press) or for Arctic benthos (Degen & Faulwetter (2019)).
        • Degen & Faulwetter (2019) The Arctic Traits Database – a repository of Arctic benthic invertebrate traits. https://doi.org/10.5194/essd-11-301-2019
      3. As I see it, as the manuscript now is, this section of the discussion lacks the results to support it.

44- Line 354-356: I would thread lightly and aim to be more specific. By rocky shores, are you referring to purely rocky substrates, or mixed ones? If purely rocky, then infauna were absent due to the nature of the substrate, and not underrepresented due to constraints on sampling.

In Barnes & Brockington (2003), their Fig. 1. Appears to represent purely rocky substrate up to 20m, without soft sediment nor mix between both (this appears deeper). This would mean that you can’t really sample infauna above 20m depth. The study of Valdivia et al. (2020) is based on epifauna only, there was no intention of sampling infauna (as I understood from the methods), and it was also only considering hard substrate, where no infauna is present. While we can argue over our reasonings based on the papers of Barnes & Brockington (2003) and Valdivia et al. (2020), it gets complicated when considering the paper of Brey & Clarke (1993). This is, mainly, because this paper is based on literature review and not on sampling per se. With that in mind, I wouldn’t consider it as a valid reference here, since there is a large mix of datasets, another thing would be if you cited specific references therein found.

Additionally, do you have infauna data in fully rocky substrates? From experience, no coring device can sample rocky substrates where no soft sediment is present, I would doubt the underwater suction sampler could outperform them (at least for hard and consolidated sediments). Different story is for mixed sediments, but this should be clearly stated.

45- Line 356-357: This is true for mixed and soft sediments, when compared with cores and grabs. For the rocky substrates is, to the best of my knowledge, within the norm.

46- Line 357-359: As I mentioned in comments 9 and 44, no functional diversity index is provided, only a number on functional groups.

47- Line 361-363: This has not been clearly showed here, and it would be good to include it in the results.

48- Line 374-378: As mentioned in comment 1 and 3, I recommend you replace the “18% more” by the actual percentage of coastline permanently ice covered.

49- Line 403: It should be “functional group diversity” rather than “functional diversity”.

50- Line 403-404: Based on comments 9, 28, 43, and 46, I think you are missing results to support this claim.

51- Line 433: I assume the “)” shouldn’t be there, or that there’s one missing before.

52- Line 445-446: I think saying 18% is “massive” is a bit exaggerated and suggest the word to be removed.

Reviewer 2 Report

this is a nice paper showing interesting and novel results relating antarctic biodiversity with carbon stocks and potential for these areas to continue storing. The study includes feedback loops with future climate change scenario and ice loss. 

Overall the study has high merits and it is well written. 

I only have a few points below:

Simple summary:

I believe this can be written in a simpler way. I personally found easier the abstract than this

Introduction

In L 61 would be important to also mention the C release mechanisms, through excretion processes and co2 release during calcification

some sentences here are also not clear. (L 61 & L 70)

L 74 the use of negative word here is perhaps confusing

Methods

The timeline could be made more clear here

2.2. 3 quadrats in each zone i assume

2.5. statistics - here I suggest expanding. In the results i find many things that are not explained here. I am not understanding what multi variables entered the multivariate analyses for example, and the log vs log graph in fig 6 is also confusing as i don't see it explained in the methods

discussion is fine but perhaps hard to read well, it could benefit from being broken down by subsections  

Round 2

Reviewer 1 Report

General comment:

The revised version of Morley et al.'s manuscript is now quite clear explaining the methods and results, which makes it easier to understand and support the discussion. This has, in turn, improved the quality of the manuscript considerable. I appreciate and thank the effort put by the authors to go through the list of comments I had during the first review round.

In it's current version, I think the manuscript should be accepted after just a few more minor (minuscule, you could say) revision, since I think there are a few small things to correct. Other than the fix in the legend of figure 5 (see comment below), most of them are mainly minor editorial fixes.

Specific comments:

  • Line 4: I guess that the "ä" should be an "m".
  • Line 58: Please replace the "Z" of zoobenthos for a "z".
  • Line 206: There's an extra space after "transformation".
  • Line 210: Please replace "route" by "root".
  • Line 240: There's an extra " ." between "shown." and "Data".
  • Line 294: There's an extra ")".
  • Line 299-300: It appears the legend of the figure was not updated, as it still mentions the the square Log10 of the number of functional groups. This should be fixed to fit the text.
  • Line 656-657: I don't know if it's a system failure or an overlook while submitting the revised manuscript, but reference 49 is included twice

Author Response

Once again we thank this reviewer for the tremendous effort they have put into improving our manuscript. We have made all the suggested changes except we were unable to find the duplicated citation "49" in our manuscript.